# Expanding the Perspective on PARP1 and Its Inhibitors in Cancer Therapy: From DNA Damage Repair to Immunomodulation

**DOI:** 10.3390/biomedicines12071617

**Published:** 2024-07-20

**Authors:** Flurina Böhi, Michael O. Hottiger

**Affiliations:** 1Department of Molecular Mechanisms of Disease, University of Zurich, 8057 Zurich, Switzerland; 2Cancer Biology Ph.D. Program, Life Science Zurich Graduate School, University of Zurich, 8057 Zurich, Switzerland

**Keywords:** PARP1, PARP inhibitors, immunomodulation, macrophages, T cells, cancer therapy

## Abstract

The emergence of PARP inhibitors as a therapeutic strategy for tumors with high genomic instability, particularly those harboring BRCA mutations, has advanced cancer treatment. However, recent advances have illuminated a multifaceted role of PARP1 beyond its canonical function in DNA damage repair. This review explores the expanding roles of PARP1, highlighting its crucial interplay with the immune system during tumorigenesis. We discuss PARP1’s immunomodulatory effects in macrophages and T cells, with a particular focus on cytokine expression. Understanding these immunomodulatory roles of PARP1 not only holds promise for enhancing the efficacy of PARP inhibitors in cancer therapy but also paves the way for novel treatment regimens targeting immune-mediated inflammatory diseases.

## 1. PARP Inhibitors in Cancer Therapy

Nearly two decades ago, the discovery of the synthetic lethality interaction between two crucial DNA damage repair proteins, PARP1 and BRCA1/2 [1,2], instigated cancer therapy with PARP inhibitors (PARPi), targeting tumors with high genomic instability (Figure 1A). In 2014, the U.S. Food and Drug Administration (FDA) approved the first PARPi, Olaparib, as a monotherapy for advanced ovarian cancer patients with germline-mutated *BRCA* [3]. Subsequent approvals of novel PARPi, such as Rucaparib, Niraparib, and Talazoparib, by the FDA and/or the European Medicine Agency (EMA) expanded the number of available options to inhibit PARP1, each distinguished by variances in their efficacy and selectivity in inhibiting PARP1 [4,5,6] due to their different binding affinities [7] and differences in their ability to trap PARP1 on chromatin [8,9]. Since then, the scope of PARPi application has broadened to the treatment of breast and pancreatic cancer patients, as well as epithelial ovarian, fallopian tube, and primary peritoneal cancer [3]. Recent FDA approvals have further extended the usage of PARPi therapy for the treatment of homologous recombination (HR)-deficient metastatic castration-resistant prostate cancer.

PARPi in clinical use primarily, but not exclusively, target the nuclear ADP-ribosyltransferase (ART) PARP1 [4], an ART that belongs to a group of structurally related proteins with closely related catalytic domains [10]. PARP1 is especially important in its role as a DNA damage sensor, as it recognizes DNA lesions and allows for the efficient recruitment of repair proteins. More specifically, DNA binding activates PARP1, resulting in the transfer of ADP-ribose onto itself and selected target proteins using NAD^+^ as a substrate. The resulting poly-ADP-ribose (PAR) chains serve as docking sites for various DNA repair factors that possess PAR-binding domains, leading to their recruitment to the DNA lesion (extensively reviewed elsewhere [11,12]. Auto-poly-ADP-ribosylation (Auto-PARylation) of PARP1 ultimately results in its release from the DNA lesion, a step that is required for reliable DNA repair [12,13]. Due to its central role in initiating the DNA damage repair cascade, PARP1 inhibition is significantly more toxic for cancer cells with a deficiency in HR compared to HR-proficient cells [1,2]. This concept of synthetic lethality appears highly promising for its apparent specificity in effectively targeting cells by exploiting a tumor-enabling and, thus, presumably essential characteristic of tumor cells [14]. However, tumors have developed mechanisms to circumvent this mode of killing, resulting in resistance [15]. The various mechanisms of PARPi resistance and potential combinatorial treatments to overcome the acquired resistance have been extensively reviewed elsewhere [16,17,18,19,20].

Besides inhibiting PARP1’s enzymatic activity, PARPi induce PARP1 chromatin retention (“trapping”) to varying degrees [8,21,22,23], which was proposed to be driven by prolonged or highly efficient catalytic inhibition of PARP1 [8,23,24], or by the inhibitors’ ability to promote a conformational change resulting in a pro-retention DNA-PARP1-inhibitor complex [9,23]. Interestingly, the reduction in PARP1’s enzymatic activity upon inhibition does not necessarily correlate with the levels of PARP1 chromatin retention [8,9,21,22,25], underscoring the idea that inhibiting PARP1’s enzymatic activity and promoting PARP1 trapping might be two independent mechanisms of inducing PARPi-dependent cytotoxicity. Over the past years, the increase in PARP1 trapping, rather than blocking its enzymatic function and thus the recruitment of DNA damage repair proteins, has been widely considered to drive cytotoxicity in PARPi-treated cells [8,9,26]. However, since recent findings suggest that the chromatin retention of PARP1 and, ultimately, cytotoxicity can be driven by either the inhibition of PARP1’s enzymatic activity or by inducing allosteric trapping of PARP1 [23], both can be considered drivers of PARPi-induced cell death. Indeed, in HR-deficient cells, PARPi-induced lethality depends on inhibiting PARP1’s enzymatic activity and thus on interfering with its role in promoting DNA integrity, while the cytotoxicity observed in HR-proficient cells stems from trapped PARP1 [21]. In light of these findings, it becomes clearer why PARPi are effective in the treatment of tumors with high genomic instability [27,28,29] while they can also induce cytotoxicity in HR-proficient tumor cells [17,30,31].

According to the human protein atlas, normal lymphoid tissues express high amounts of *PARP1* mRNA and PARP1 protein compared to other tissues [32], which implies a functional contribution of PARP1 in this compartment and potentially broader applicability of PARPi. The notion of PARP1 playing a role in cellular function beyond its role in DNA damage repair is underscored by a genome-wide CRISPR loss-of-function screen of over 1000 cell lines (https://depmap.org/portal/, accessed on 9 July 2024, including only those tumor lineages with at least 10 characterized cell lines) that suggests that PARP1 dependency across different tumor types is relatively comparable (Figure 2A). Moreover, the correlation between PARP1 gene dependency and the sensitivity to the PARPi Talazoparib is very weak in most of the publicly available cancer cell lines (Figure 2B). Despite Talazoparib’s high efficacy in inhibiting PARP1’s enzymatic activity and inducing effective PARP1 trapping [23], these data highlight the discrepancy between PARP1’s enzymatic activity and the mere presence of PARP1. They also suggest additional roles of PARP1 that may be independent of ADP-ribosylation and beyond PARP1’s role in DNA damage response pathways.

## 2. Revising PARP1 Inhibitors: Roles beyond Their Canonical Function in DNA Damage Repair?

The human protein atlas clusters genes according to their mRNA expression. Interestingly, according to the *PARP1* expression pattern, it belongs to the “Lymphoid tissue—Cytokine signaling” cluster, along with numerous cytokines and chemokines, as well as chemokine receptors and leukocyte cell surface molecules [32]. Indeed, one important emerging function of PARP1 is its role in modulating cytokine and chemokine expression [33,34,35].

The vital role of both adaptive and innate immune cells in mediating PARPi-induced anti-tumor immunity has been demonstrated in murine tumor models [36,37,38]. For example, the ADP-ribosylation of cGAS has been proposed to inhibit its function in inflammatory cytokine expression (Figure 1B) [39]. In addition, Olaparib treatment has been shown to enhance cGAS/STING signaling in tumor cells and dendritic cells through the increase in PARPi-induced cytosolic DNA in tumor cells [37,38]. The resulting increase in cytokine expression is correlated with enhanced T cell tumor infiltration and improved tumor control [37,38]. In contrast to this indirect effect of PARPi [36], direct PARP inhibition-mediated immunomodulatory effects in immune cells add an additional layer of complexity in the context of tumors. Considering the delicate equilibrium between tumor-promoting inflammation and anti-cancer immunity [40], it seems highly important to understand, evaluate, and consider both the primary, direct, and secondary, indirect systemic effects of PARPi on anti-tumor immunity. Could these effects potentially be harnessed or warrant consideration in the context of patient treatment strategies?

## 3. Transcriptional Co-Activator PARP1 Promotes Cytokine Expression in Macrophages

While most studies focus on the immunomodulatory effect of PARPi as a secondary response to the PARPi effects in cancer cells, Olaparib has been demonstrated to promote the anti-tumorigenic function of macrophages directly [36]. Importantly, the augmentation of anti-tumorigenic macrophage activity following high-dose Olaparib treatment was attributed to elevated levels of NAD^+^ and reactive oxygen species (ROS) rather than the PARP1-dependent ADP-ribosylation of target molecules and subsequent alterations in cellular signaling, which were not investigated further (Figure 1C) [36]. Although the observed effects in high-dose Olaparib-treated macrophages stem from a global cellular increase in NAD^+^ rather than targeted ADP-ribosylation, nonetheless, it remains possible that PARP1-dependent ADP-ribosylation exerts its function beyond DNA damage repair.

PARPi treatment has proven to be beneficial when administered in mouse models of inflammatory disease and correlated with a decrease in cytokines, chemokines, and adhesion molecules such as TNF, IL-1β, and CCL2 [33,41,42,43]. Numerous pathways rely on the presence of PARP1 or its enzymatic activity for effective signaling [44]. One extensively studied mechanism of how PARP1 augments cytokine expression involves the regulation of NF-κB signaling (Figure 1D). Both PARP1 protein and its enzymatic activity have been associated with increased NF-κB signaling [45,46]. The activation of NF-κB is facilitated through the interaction between PARP1 and transcriptional co-activators CBP and p300. This interaction leads to the acetylation of PARP1, which enables the complex to engage with the p50 subunit of NF-κB and initiates the transcription of pro-inflammatory cytokines [45,47]. Indeed, in the absence of PARP1, a decrease in NF-κB target gene expression was observed in vitro [43,48]. Furthermore, PARP1’s enzymatic activity appears to be important for NF-κB signaling. The ADP-ribosylation of p65 was proposed to be critical for its prolonged retention in the nucleus and ultimately enhancing NF-κB-dependent gene expression by reducing its interaction with the nuclear exporter Crm1 [46]. Additionally, DNA damage induced by irradiation and the consequent activation of PARP1 was shown to drive NF-κB activation [49]. Besides promoting NF-κB activity, LPS-induced PARP1 activation has also been reported to result in histone modification and histone destabilization, which in turn enhances the accessibility of NF-κB binding sites [50]. The importance of PARP1-dependent cytokine expression was demonstrated in animals that lack PARP1 and thus exhibit resistance to LPS-induced endotoxic shock [48]. This resistance is associated with reduced NF-κB activation and lower levels of cytokines such as TNF⍺ and IFNɣ [48]. In myeloid-specific PARP1 knockout (KO) animals, LPS-induced expression of IL12 and IL18 in macrophages was shown to depend on PARP1 but not its enzymatic activity [51]. In contrast, another study showed that, independently of the type of stimulus, both PARP1 KO and PARP1 inhibition led to a reduction in numerous cytokines [52].

Besides NF-κB, the interplay of PARP1 with the transcription factors NFAT and STAT1 was described to play a role in macrophages [53,54]. While the LPS-induced ADP-ribosylation of NFAT in macrophages has been described to enhance DNA binding and target cytokine expression in macrophages [54], cytokine expression in IFNɣ-stimulated macrophages is governed by ADP-ribosylated STAT1 (Figure 1E,F) [53]. The ADP-ribosylation of two distinct sites on STAT1⍺ has been reported to be essential for its binding to STAT1-specific DNA motifs and for the functional interaction of PARP1 with the transcriptional co-activator p300 [53]. PARP1 inhibition led to a reduction in IFNɣ-induced chemokine expression in induced bone marrow-derived macrophages (iBMDMs), and the stimulation of ADP-ribosylation-deficient STAT1⍺ iBMDMs resulted in the impaired induction of genes involved in the innate immune response [53].

## 4. T Cell Anti-Tumor Immunity: Synergistic Anti-Tumor Effects of PARP Inhibitors and Immune Checkpoint Inhibitors

The depletion of T cells and macrophages in PARPi-treated animals was shown to correlate with an increase in tumor volume or a decrease in survival, respectively [36,37], underscoring the importance of effective T cell and macrophage immunity in tumor control. Harnessing the body’s immune response against tumors with immune checkpoint inhibitors has become a widely used therapy in many cancers [55]. However, for this promising treatment approach, the occurrence of resistance and non-responders has also become evident [55]. Interestingly, the combination of immune checkpoint inhibitors with PARPi administration has been shown to significantly reduce the tumor burden [38]. Furthermore, the expression of PD-L1 in cancer cells has been shown to be negatively regulated by PARP1, and conversely, PARP1 inhibition leads to an increase in PD-L1 expression, proposing one possible explanation for the more effective control of tumor growth with the combinatorial treatment compared to PARPi monotherapy [56]. Thus, PARPi potentiate the efficacy of immune checkpoint inhibitors by creating an optimal environment via the induction of DNA damage and cytoplasmic DNA, the latter of which acts as a potent immune stimulus [37,38,57]. Indeed, the ability of PARP inhibitors to promote an anti-tumorigenic environment through chemokine release, which favors T cell infiltration while suppressing the migration of myeloid-derived suppressor cells, may explain why PARPi administration and CAR T cell therapy have been shown to be synergistic in preclinical studies [58,59].

As is the case for macrophages, PARPi treatment also directly modulates T cell function by influencing signaling pathways. While the ADP-ribosylation of the transcription factors NFAT1 and NFAT2 in T cells enhances the nuclear export of NFAT, PARPi promote NFAT-dependent gene transcription (Figure 1G) [60]. Interestingly, the stimulation of T cells in the absence of PARP1 resulted in more down- than upregulated genes, with the dampened genes potentially representing NF-κB-regulated genes [61]. Among the downregulated genes were those encoding for Th2 cytokines IL4 and IL5, demonstrating impaired Th2 cell polarization in the absence of PARP1 [61,62]. A role for PARP1 in the Th2 axis is further supported by the significantly decreased STAT6 and Gata3 levels, as well as by the decrease in Th2 cytokines such as IL4, IL5, and IL13, in stimulated PARP1 KO or PARPi-treated animals and splenocytes [62,63,64,65].

Contradicting reports exist regarding the effect of PARPi on the Th1 subset. Typical Th1 cytokines were reported to be unaffected or upregulated upon PARP1 inhibition or PARP1 KO [61,62,64]. Other reports indicate that PARPi treatment resulted in the downregulation of Th1 cytokines such as IFNɣ [66,67,68], suggesting stimulation- and disease model-dependent effects of PARPi. Which cytokines are expressed by T cells depends on the type of T helper subset that the T cells differentiate into [69]. Thus, besides a direct influence of PARP1 on T cell cytokine transcription, this contradiction could also be explained by PARP1’s influence on the proportion of different T helper cell subsets [35,67,70,71,72]. It is therefore not surprising that PARP1-deficient mice were also reported to efficiently control the cancer growth of a breast tumor cell line [73].

Considering that depending on the PARPi concentration used, cellular NAD^+^ can be significantly elevated [36], thereby impacting the anti-tumor function of both macrophages and T cells [36,74], the highly context-dependent and stimulus-specific transcriptional effects of ADP-ribosylation inhibition across various immune cell types is evident. This complexity suggests that the effect of PARPi, beyond their synthetic lethality interaction in HR-deficient cancer cells, requires more investigation. The therapeutic outcomes of PARP inhibition may be profoundly influenced by the tumor microenvironment and the immune system overall in the specific disease.

## 5. Outlook

While there is a strong consensus on the synergistic effects of PARPi and immune checkpoint blockade, the prevailing understanding and assumption about the combinatorial impact of this treatment remain predominantly DNA damage- and cancer cell-centric. PARPi-induced DNA damage and subsequent cell death have been shown to enhance the efficacy of immune checkpoint inhibitors by increasing the release of immunogenic material and fostering a pro-inflammatory, anti-tumorigenic microenvironment [75,76,77]. Conversely, since PARP1 inhibition leads to the upregulation of PD-L1, the co-administration of immune checkpoint inhibitors and PARPi has been observed to be beneficial (Figure 1H) [75,76].

However, a purely cancer-centric view of PARPi may overlook the broader, systemic impacts that this treatment might have. The appropriate activation of both T cells [78,79] and macrophages [80] is crucial for robust and functional anti-tumor immunity. The role of PARP1 as a co-translational regulator of cytokine expression and in the polarization of T cells and macrophages has been extensively discussed previously. Given PARP1’s critical function in cells that mediate anti-tumor immunity, along with reported adverse effects of PARPi treatment such as hematological toxicities [81,82], it seems to be imperative to investigate the direct and immediate effect of PARP inhibitors on immune cells within the context of cancer therapy in greater detail.

In light of the broad effects of PARPi on different cell types and since, at present, the experimentally proven beneficial effect of PARPi treatment seems to rely mostly on their impact on cancer cells, the localized administration or release of PARPi might be advantageous in terms of reducing side effects like hematological toxicity [83,84]. In addition, this targeted approach could also enhance cancer treatment efficacy [85], with a PARPi-driven increase in cytosolic DNA, DAMPs, and neoantigens only in those cells specifically that are cancerous. For example, the nanoparticle delivery of Talazoparib has been proven to be a promising method for enhanced cancer control, increasing tumor infiltration by immune cells, including an increase in intratumoral T cells, while simultaneously correlating with a decrease in the percentage of myeloid-derived suppressor cells [86]. This tumor targeting of PARPi could indeed limit the adverse effects on immune cells while simultaneously promoting macrophage-dependent anti-tumor cytotoxicity via elevation of ROS levels [36]. Comparing local versus systemic PARPi administration could provide insight into the optimal root of delivery, enabling detailed in vivo profiling of the activation status of tumor-infiltrating and circulating immune cells for the development of more effective treatment strategies. Beyond the spatial considerations of PARPi treatment, it would be interesting to contemplate the timing of administration. Could the effects of PARP inhibition on cancer cell genetic integrity emerge more swiftly than its immunomodulatory impact? Or, might the effect on immune cells be more transient and rapidly reversible upon withdrawal of the inhibitor?

As described above, the immune-regulating effects of PARP1 have been extensively studied at the cellular level. Additionally, in mouse models of immune-mediated diseases, PARPi administration has been shown to alter the levels of various inflammatory cytokines in the blood, underscoring the role of PARP1 in the immune response without identifying specific PARPi-sensitive cell types [33,34]. Thus, PARPi administration may have much broader therapeutic applications than currently utilized. Systemic PARP inhibition could potentially serve as a non-specific anti-inflammatory agent in diseases characterized by high and dysregulated cytokine expression. Thus, further comprehensive research into the local and systemic effects of PARPi could unveil new therapeutic avenues, ultimately refining cancer treatments and extending their benefits to a broader range of immune-mediated conditions.

While extensive research in recent years has primarily focused on PARP1 and its immunoregulatory effects, the spotlight is increasingly turning to other ARTs that play important roles in modulating immune responses and may offer new avenues for cancer therapy. The clinical potential of inhibiting ARTs besides PARP1 is exemplified by a PARP7 inhibitor in phase I clinical trials [87]. In preclinical studies, the inhibition of PARP7 has shown efficacy in reducing tumor growth through both cell-autonomous mechanisms and by enhancing type I interferon (IFN) signaling and anti-tumor immune responses [88,89,90]. Furthermore, the anti-tumorigenic effects of PARP7 inhibitors have shown a strong synergy with anti-PD-1 treatment [88]. Similarly, the inhibition of PARP14 has been shown to synergize with anti-PD-1 therapy, improving survival in tumor-bearing mice by enhancing anti-tumor immune responses [91]. The depletion of PARP11 has been reported to prevent the PARP11-dependent loss of the type I interferon receptor IFNAR1 in T cells, thereby promoting the anti-tumorigenic functions of cytotoxic T lymphocytes. Additionally, PARP11 depletion in CAR T cells has been shown to increase IFNAR1 levels and enhance their efficacy in controlling tumor growth [92]. Taken together, this emergence of PARP1 and other members of the ART family in the regulation of immune pathways underscores the immense potential for the further exploitation of these enzymes not only in cancer therapy but also in the treatment of immune-mediated diseases.

## Figures and Tables

**Figure 1 biomedicines-12-01617-f001:**
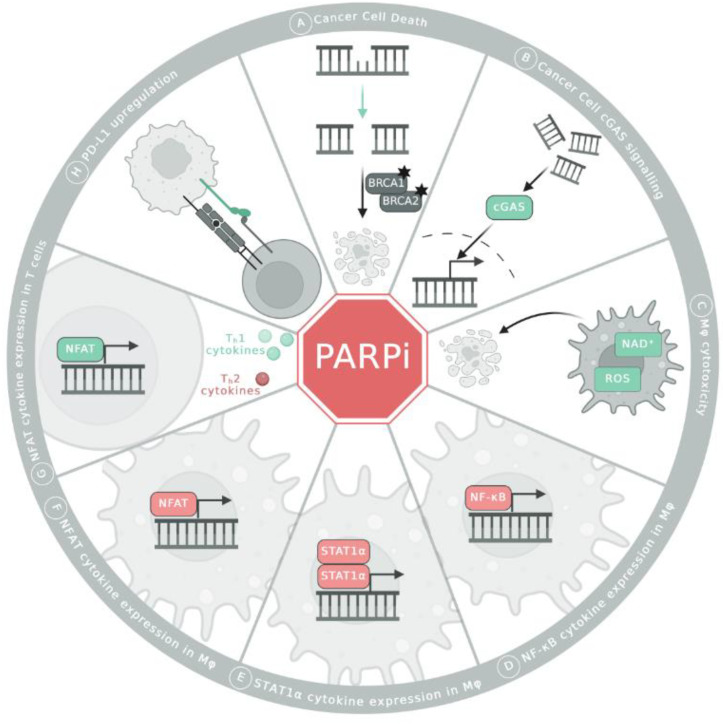
PARPi effect in cancer cells and immune cells. Depicted in green are effects that are promoted by PARPi treatment, and effects that are inhibited by PARPi treatment are shown in red. (**A**) Synthetic lethality interaction of homologous recombination (HR)-deficient cancer cells and PARPi treatment leads to cancer cell death. (**B**) PARPi treatment also inhibits ADP-ribosylation of cGAS in cancer cells, thereby enhancing the cGAS/STING signaling pathway. (**C**) In macrophages, PARPi elevate levels of NAD^+^ and ROS. This shift in metabolism enhances the anti-tumorigenic function of macrophages. (**D**–**G**) PARPi up- and downregulate the transcription of various cytokines in both macrophages and T cells. (**H**) In cancer cells, PARPi treatment upregulates PD-L1 expression, which may have implications for the interaction between cancer cells and the immune system.

**Figure 2 biomedicines-12-01617-f002:**
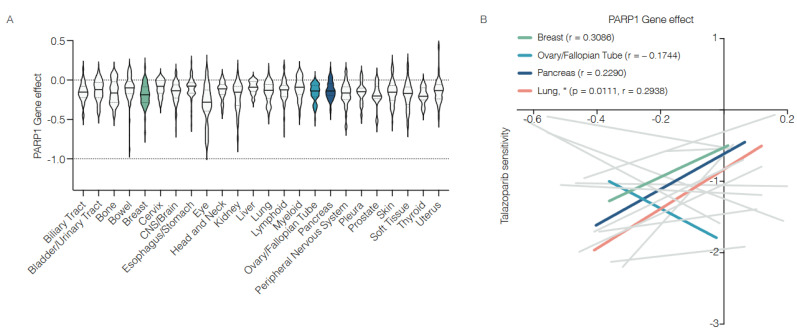
Comparable PARP1 gene effects across tumor cell lines. To compare the effect of PARP1 and its enzymatic activity across different cell lines, we used publicly available datasets of genome-wide CRISPR loss-of-function and drug sensitivity screens. These datasets provide scores for gene dependency and drug sensitivity (https://depmap.org/portal/, accessed on 9 July 2024). (**A**) Dependency on PARP1 in several of the examined cell lines. Colored groups represent cell lines derived from breast, ovarian, fallopian tube, and pancreas tumors, all of which are treated with PARPi in clinics. (**B**) Correlation plot of PARP1 gene dependency scores (*x*-axis) vs. Talazoparib sensitivity scores (*y*-axis). Except for lung cancer cell lines (colored in red), which show a significant correlation between PARP1 gene effect and PARPi treatment efficacy, most cancer cell lines, including those derived from breast, ovarian, fallopian tube and pancreas tumors (colored in green/light blue/ dark blue), exhibit a low and non-significant correlation between PARP1 gene effect and PARPi efficacy. The cancer types shown in gray are those for which data on the PARP1 gene dependency score (Figure 2A) or data on the PARP1 gene dependency score and on the Talazoparib sensitivity score (Figure 2B) were available for at least 10 cell lines. In none of these cancer types was a strong or significant correlation found between PARP1 gene effect and PARP1 inhibition. * = *p* ≤ 0.05.

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
