# Peer review of "Expanding the Perspective on PARP1 and Its Inhibitors in Cancer Therapy: From DNA Damage Repair to Immunomodulation"

_biomedicines, 2024, doi:10.3390/biomedicines12071617_

Round 1

Reviewer 1 Report

Comments and Suggestions for Authors

In this review, authors  move from the canonical function of PARP1 in DNA damage repair to the expanding roles of PARP1 interplayed with the immune system during tumorigenesis. The review is a comprehensive collection of the literature regarding the exploitation of PARP inhibitors (PARPi) in cancer therapy, with a particular focus on PARPi immunomodulatory roles.

In the initial part, authors reports the canonical exploitation of PARPi as a therapeutic strategy for tumors with high genomic instability, particularly those harboring BRCA mutations. Authors describes well how cytotoxicity can be driven by either the inhibition of PARP1 enzymatic activity or by inducing allosteric trapping of PARP1. Then, authors focus on the immunomodulatory role of PARPi (see strengths)

The review is written in a linear way and authors perspectives are clear. Unfortunately  authors have paid little attention to the figures and figure legends.

This review can be accepted for publication, only if the weak points  will be addressed.

Strenghts

1.      This review highlights the importance of  using PARPi in cancer therapy focusing on the immunomodulatory effect of PARPi

2.      Authors focus on the direct effects of PARPi treatment on macrophages and T cell function by modulating signaling pathways:

a.         On macrophages - PARPi, through the regulation of NF-κB signalling, cause a decrease in cytokines, chemokines, and adhesion molecules such as TNF, IL-1β, CCL2; through the  transcription factors NFAT and STAT1 controls  cytokine expression and improve induction of genes involved in the innate immune response.

b.         On T cells - PARPi promote NFAT-dependent gene transcription, up- and down-regulating the transcription of various cytokines.

As reported, contradicting reports exist regarding the effect of PARPi on the Th1 subset and the response depends from the tumor microenvironment and the immune system overall in the specific disease.

3.      In the last paragraph, authors dwell to illustrate the beneficial effects of the combination of immune checkpoint inhibitors and PARPi in tumour treatment, since the efficacy of immune checkponts inhibitors is increased both by PARPi-induced DNA damage and the PARPi-induced upregulation of PD-L1

4.      Authors state that the beneficial effect of PARPi treatment seems to rely mostly on their effect on cancer cell and suggest that  the localized administration or release of PARP might be advantageous in terms of reducing side effects like hematological toxicity.

5.      Authors acknowledge the existence of  problems on the localized and systemic effects of PARPi, due to the complexity of the tumor system, although they predict a potential for new therapeutic avenues in cancer treatments and immune-mediated conditions.

Weaknesses

Figures and figure legends should be improved to help readers to follow the rationale of the speech and to summarize the content from different research articles.

Authors should change the figure numbering. The first figure cited in the text is figure 1 (and not 2). The colours must be shown  where indicated

Figure 1 (change in Figure 2)

I would indicate more clearly the roles played by PARP inhibitors in antitumor immunity, referring with alphabetical letters or numbers to the corresponding section of the figure illustrated by authors.

For example when authors report in the text that the” Indirect effect Olaparib treatment is to enhance cGAS/STING signaling in tumor cells and dendritic cells through the increase of PARPi-induced cytosolic DNA and  increase in cytokine expression (lane 94-97)”, I would cite Fig. 1 B in the text and add a reference element in figure.

Figure2  (change in Figure 1)

Authors show genome-wide CRISPR loss-of-function screen of over 1000 cell lines to suggest that different tumor types are dependent on PARP1. It is reported in the actual Figure 2A:” Represented in green are cell lines derived from breast, ovarian, fallopian tube, and pancreas tumors,, all of which are treated with PARPi in the clinics. They are not represented in green

Figure 2B What type of scale was used for the graph of correlation?

Usually the correlation coefficient is measured on a scale that varies from + 1 through 0 to – 1. Complete correlation between two variables is expressed by either + 1 or -1.

Which cell line do the grey lines indicated on the graph refer to? There is no clear indication

Author Response

Please find the Response in the attached word file.

Reviewer 2 Report

Comments and Suggestions for Authors

In the paper titled: "Expanding the Perspective on PARP1 and Its Inhibitors in Cancer Therapy: From DNA Damage Repair to Immunomodulation" by Flurina Bohi and Micheal O. Hottiger, submitted to Biomedicines MDPI, authors review literature regarding PARP inhibitors mechanism of action in cancer in context of immunomodulation.

Throughout the text, although authors focus on T cells, however there is no mention of chimeric antigen receptor modified T cells (CAR-T) cell therapy (immunotherapy) in combination with PARP inhibitors. Please provide additional couple of sentences, how CAR-T cell therapy and PARP inhibitors are being used in context of drug synergy between PARPi and PD-L targeted therapies even if it is just in pre-clinical studies. The idea of combining drugs that target immune system with PARP inhibitors needs to be clearly stated, and developed, supported with appropriate references.

Figure 2 is a bioinformatic analysis that is not clear; please clarify the main message for the analysis of selected cancer cell lines in context of PARP gene effects. Please be concise and focused while describing PARP dependency in different tumor cell lines, especially focusing on developed target tumor cell lines where PARPi are used in the clinic. In Panel B, please change the colors of ovarian cell line from green to a different unique color to see differences/similarities between breast and fallopian tube/ovary.

It is recommended to include additional illustration showing differences in PARP inhibition in macrophages vs dendritic cells vs T cells (including CAR-Ts) in tumor stroma. This information should be clearly stated in the text and illustrated. Additional Figure with more details than shown in Figure 1, would support PARP inhibitors immunomodulation and show the way to use drug combinations with existing immunotherapies

Author Response

Please find our reply to the reviewers comments in the attached word file. 

Reviewer 3 Report

Comments and Suggestions for Authors

This is a manuscript from one of the leaders of the ADP-ribosylation field co-authored by one of his colleagues. Therefore, the MS is authoritative, informative, accurate and highly timely.

I only have a few minor remarks as follows:

1. In line 11-12, there is an accidental termination of the line separating the word PARP1 to PAR and P1

2. In lines 48 and 50, I suggest changing "resistances" to "resistance"

3. In line 112, I suggest slightly rephrasing „…which not further investigated.”  to „which was not investigated further”.

4. In line 324, there is a duplication in the title of the cited reference „PARP-inhibition reprograms macrophages toward an antitumoranti-tumor phenotype”

5. In line 227, I suggest rephrasing: „Comparing localized versus systemically PARPi administration” to "Comparing topical versus systemic PARPi administration"

6. Although the MS focuses on PARP1 and its inhibitors in cancer immunomodulation, I suggest briefly covering other immunomodulatory PARPs with special regard to PARP14. (Perhaps a short paragraph on this topic in the "Outlook" section might be worthwile.)

Author Response

(The authors gave the same response as above.)
